# Increased Cattle Feeding Precision from Automatic Feeding Systems: Considerations on Technology Spread and Farm Level Perceived Advantages in Italy

**DOI:** 10.3390/ani13213382

**Published:** 2023-10-31

**Authors:** Elio Romano, Massimo Brambilla, Maurizio Cutini, Simone Giovinazzo, Andrea Lazzari, Aldo Calcante, Francesco Maria Tangorra, Paolo Rossi, Ambra Motta, Carlo Bisaglia, Andrea Bragaglio

**Affiliations:** 1Council for Agricultural Research and Economics (CREA), Research Centre for Engineering and Agro-Food Processing, Via Milano 43, 24047 Treviglio, Italy; elio.romano@crea.gov.it (E.R.); maurizio.cutini@crea.gov.it (M.C.); simone.giovinazzo@crea.gov.it (S.G.); andrea.lazzari@crea.gov.it (A.L.); carlo.bisaglia@crea.gov.it (C.B.); andrea.bragaglio@crea.gov.it (A.B.); 2Department of Agricultural and Environmental Sciences-Production Territory Agroenergy, Università Degli Studi di Milano, Via Celoria 2, 20133 Milan, Italy; aldo.calcante@unimi.it; 3Department of Veterinary Medicine and Animal Sciences, Università Degli Studi di Milano, Via Dell’Università, 6, 26900 Lodi, Italy; francesco.tangorra@unimi.it; 4Fondazione CRPA Studi Ricerche, V.le Timavo, 43/2, 42121 Reggio Emilia, Italy; p.rossi@crpa.it (P.R.); a.motta@crpa.it (A.M.)

**Keywords:** total mixed ration, TMR, AFS, automation, beef, dairy, ruminants

## Abstract

**Simple Summary:**

Livestock farming is experiencing growing levels of automation due to the high number of repetitive tasks requiring little skill and high workloads. Automatic feeding systems for ruminants represent a valuable option for farmers to reduce their daily workload and increase animal welfare and production quality. The farmers’ perspective matched the outcome of the studies on the benefits of automatic feeding system adoption (lower energy requirements, lower feed waste, and increased animal welfare). However, such benefits only exist if thorough economic and structural planning considers all of the farming site’s features to exploit the feeding robots’ flexibility at its best. We suggest that automatic feeding systems may be fundamental to the involvement of younger generations in animal farming and increasing the inclusivity of such an activity with the subsequent fostering of female entrepreneurship, always complying with good feeding practices.

**Abstract:**

Automation reduces the impact of farming on climate change and helps farmers adapt to its financial impact. Automatic feeding systems (AFSs) increase the ruminant’s feeding precision and ease operators’ workload. Such systems exist on a spectrum, requiring varied levels of operator support and installation complexity. A recent survey on farmers pointed out that those already running an AFS and those willing to buy one appreciate its ease of use, the resulting animal welfare, and the resulting overall benefits (increased production, farm profitability, and reduced feed waste). Whether technologically complex or not, studies have confirmed the benefits that farmers perceive to be underlying the remarkable reduction in the environmental impact of feeding operations (AFSs are electrically powered), the increase in animal welfare resulting from reduced conflicts around accessing the feed bunk, and the constant availability of fresh feed. However, their introduction should follow accurate and holistic structural and economic planning for existing and newly built facilities. The availability of public subsidies plays an essential role in pushing farmers to adopt the most modern digital technologies, whose benefits may even increase when farmers couple them with interconnected sensors to monitor animals’ physiological states.

## 1. Introduction

Automatic feeding systems (AFSs) ease the workload of dairy farmers, save time, and increase workload flexibility.

Decreased workloads related to shed procedures allow the farmers to focus on organizational activities. As reported by Hansen et al. [1], who studied the farmers’ well-being following the adoption of an automatic milking system (AMS), automatic systems facilitate the occurrence of these scenarios. Additionally, a review conducted by Lovarelli et al. [2] highlights how automatic systems (AFSs and AMSs) and precision livestock farming (PLF) can improve the economic sustainability of livestock systems. A comparison of a conventional dairy farm with a precision-agriculture-based farm by Bragaglio et al. [3] found that using an AFS and AMS led the precision-agriculture-based farm to lower its values of MJ consumption and oil equivalents, which resulted in a positive impact on the environment. In terms of the effect of feeding systems on ammonia and greenhouse gas (GHG) emissions from the decomposition of dairy cows’ excreta on the housing floor, Rodrigues et al. [4] have pointed out that administering the total mixed ration (TMR) with an AFS can lower N–NH_3_ emissions (g/kg excreta) compared to conventional feeding, without a significant difference regarding their contribution to GHG emissions.

The fully functioning AFS relies on the studies that introduced the concept of total mixed ration (TMR) for ruminants’ feeding. The presentation of the TMR feeding concept for dairy herds dates back to the 2nd half of the 20th century [5] following the increases in herd dimension and the modernization of animal housing (i.e., free-stall handling and the use of a milking parlor), which resulted in a significant rise in milk production [6].

However, the introduction of TMR rationing in dairy and beef farms had to wait until the 1960s [7], thanks to the design, development, and subsequent spread of mechanical mixing devices (stationary mixing plants and mixing feeding wagons—MFWs), which have become the dominant equipment for TMR preparation and distribution on cattle farms [8]. TMR feeding has proven to be the feeding method closest to reared ruminants’ physiology [9,10], obtaining remarkable consensus given the suitability of existing AFSs for European dairy farms [11].

The TMR fosters ruminants’ feed reception with subsequent improvements in their efficiency and health status [12] provided operators adopt proper precautions (e.g., water addition during mixing) and adequate mixing times [13,14,15], which are fundamental regarding the TMR particle size and consistency limiting aspects [16,17].

### 1.1. The TMR–Mixer Interaction: Consistency Matters

Mixers should uniformly blend particles of different sizes, moisture content, and bulk density. Ideal mixing is the state in which any sample removed from the TMR has the same composition so that any animal taking a mouthful of feed receives a homogeneous sample of the combined ingredients. The mixed ration should mask less palatable feeds to prevent animals from sorting the ration [18]. However, the feed ingredients themselves hinder the mixer’s potential for a perfectly homogeneous ration. In addition, the mechanical forces that mix the ration can also cause particle size reduction, which may somehow represent a side effect of the mixing. Therefore, controlling TMR physical variability (in time and space) is essential.

On the one hand, inadequate TMR particle size causes the ruminal acetate-to-propionate ratio to decrease with a subsequent reduction in the ruminal pH [19,20], and may also result in animal health problems [21,22]. On the other, the unavoidably occurring daily changes in dry matter (DM) concentration and nutrient composition of ration components (e.g., forages) make the fed ration different from the formulated one, causing rearing feed costs and environmental impact to rise [23]. Costa et al. [24] highlighted DM and neutral detergent fiber (NDF) differences along the feeding alley, revealing the presence of inconsistent TMR composition for the various cows already at the distribution, even when the sampling at the wagon level after the cutting–mixing procedure showed good ration uniformity.

### 1.2. Towards Increasing Automation Levels

Besides the ration consistency issues (also resulting from operator errors in the loading of the mixer), the workload of dairy farming also requires attention. According to the European Working Conditions Survey 2015 [25], agriculture is among the sectors in which full-time employees work a high number of hours (on average 40.5) per week (no matter the weekends or holidays), and a more recent survey has shown low scores of agriculture in quality assessments related to the physical work environment and working time [26]. Specifically in dairy farming, farmers perceive repetitive tasks like feed distribution and cubicle maintenance as physically very strenuous, albeit required for animal tending [27]. Such a heavy workload caused Finnish dairy farmers to undergo substantial social distress [28], underlining automation’s essential role in this context. Höhendinger et al. [29], focusing on the evolution of West German family-run dairy farms, pointed out the increasing importance of routine task automation in dairy farming, which causes the total working time requirements to decrease, with the most considerable savings occurring in the milking and feeding processes. The Food and Agriculture Organization of the United Nations [30] has underlined the existence of technologies whose modularity of development enables the matching of the needs of farms of different sizes and income levels. Such technologies can automate, in part or in whole, the rationing process, mitigating the abovementioned limiting aspects of TMR. NIR sensors, allowing the non-destructive and real-time monitoring of the components of the ration, are helpful to operators in reducing the effect of seasonality, reducing the gap between the fed and the formulated ration as much as possible [31,32,33,34]. Image analysis techniques have recently allowed dairy farmers to monitor the fibers’ homogeneity and length inside the feed mixer during TMR preparation [35,36]. Advances in animal science and their related technologies have improved the performance of reared animals regarding healthy production, environmental protection, and socio-economic aspects [37,38]. Among these, workload quality and quantity are fundamental; the appropriate inclusion of innovative technology allows better farm labor management [39], although the workload for staff training and herd monitoring may initially be increased [40]. The use of AFSs has many advantages, including promoting women’s entrepreneurship in agriculture because AFSs can help balance work and family responsibilities, improving women’s prospects. While there are driving expectations of achieving success [41], women entrepreneurs often experience work–family conflict despite appreciating work autonomy and flexibility [42,43]. Therefore, AFSs can be a viable solution to meet the employment and career needs of many women in agriculture [44].

In light of the productive and social role of animal husbandry [45,46], the great affinity that automatic systems for the distribution of the ration have for the needs of farmed animals assumes not only technical–professional value but also retraining and reorganization in the world of work in agriculture. The machines and devices used in the barn must also become autonomous to adjust to the information more quickly and efficiently without putting additional strain on the farmer [47]. In principle, an AFS can be a good opportunity for optimizing working time and workload in dairy farming [48,49], following the general trends driving automation on the farm [50].

Although a TMR formulation may be technically correct, many variation sources influence the ration that cows consume [51]. Operating an AFS for TMR preparation delivery requires the farmer to increase the daily consistency of operation with a resulting mixing protocol (i.e., the fill order of the mixer, the mixing time, and the moisture levels of the feedstuffs) that optimizes the preparation of the TMR [52] and increases the precision of feeding as it reduces the difference between the animal nutritionist’s planned ration and that currently delivered.

### 1.3. Aim of the Review

This review regards the origin and the technology that led to the development of automatic feeding systems (AFSs), focusing on the technology that Italian farmers have chosen and deepening the reasons for their choice, which rely on the advantages farmers perceive. The study also tackles animal welfare issues and the economic and planning requirements resulting from their purchase to draw future recommendations for their installation.

## 2. AFP and AFS Technological Solutions

Feeding cattle on livestock farms is a labor-intensive operation. AFPs are the first tools developed for the recurring pushing and approaching of the feed that the animals move away from the manger when they sort the TMR against long forage particles in favor of the smaller, high-starch grain particles [53]. AFPs warrant constant cattle feed availability in the manger between TMR distributions.

AFS design aims to reduce workforce use and improve work flexibility. Their introduction dates back to the end of the 20th and the beginning of the 21st century, first in the Northern European countries and then in Italy (since 2013, mainly in the Northern Italian regions) [54]. However, their versatility has caused them to go far beyond the initial concept, making them complementary to the well-consolidated chopping–mixing wagon technology in both milk and meat production and a wide range of farming situations (e.g., farms in mountainous areas or aimed at the protected designation of origin productions) [55,56]. The systems currently on offer differ in their mode of operation and increasing complexity of technological level; ration preparation and delivery may be carried out in a stationary mixer, which feeds ration-delivering wagons, up to fully automatic systems that carry out all of the operations necessary to prepare the TMR autonomously.

### 2.1. Automatic Feed Pushers

As mentioned above, during feeding, animals carry out a feed-sorting action of the TMR [53], which results in the need to repeatedly bring the TMR closer to the feeding bunk throughout the day [57,58]. Such an action is usually carried out manually or using specific machinery; however, AFPs allow for its automation, representing a first step towards the automation of the barn. They push feed closer to the feed bunk many times a day, with numerous benefits to cattle and farmers [58]. Nabokov et al. [59] pointed out that the animals’ regular feed ingestion resulting from AFP adoption increases the farm’s production potential, which also causes a high return on investment (87.8%) and a payback period of 407 days. However, Barrett and Dahl [58] reported a wide range of payback times when referring to the full repayment of AFP investment; in their work, they considered the labor savings (based on a pushing frequency of 2–4 times per day) excluding the other benefits of the feed pushers. However, even though feed push-ups between feed deliveries ensure continuous access to feed, they have been proven to lack the stimulating effect on feeding activity provoked by the delivery of fresh feed [60,61].

The market offers various AFPs; the current bibliography does not present any classification. However, it is possible to divide this technology into two categories: automatic-guided feed pushers and automatic self-propelled feed pushers. The former is bound to a track providing them trajectory and energy; the latter runs on batteries and follows routes made by arrays of magnets or metal strips inserted in or above the paving material of the feeding alleys.

Figure 1 reports the market offer of AFPs, the results of which mainly emphasize self-propelled ones (86%), followed by those guided by tracks (14%). Self-propelled robots were further divided based on how they push the forage: self-propelled with a rotary tambour (58%), self-propelled with an auger (21%), and other types of self-propulsion (7%). Recent advances have introduced the upgrading of AFPs with a controlled dispenser of feed additives to facilitate the feeding process and optimize the dosing of concentrated additives [62].

### 2.2. The Current Situation

Nowadays, the AFS market is thriving. In 2018, Tangorra et al. [64] identified 20 AFS manufacturers and more than 1250 robotic units on farms. Reger et al. [65] recently listed 24 AFS-producing companies, including those operating under third-party brands. The AFS types currently on the market refer to three increasing automation levels [66]:Stage I: machinery whose automation refers specifically to TMR shredding, mixing, and distribution (several times a day). They commonly rely on a stationary mixer that operators fill daily with each ingredient of the TMR from bunker silos either manually or using suitable loading machines.Stage II: compared to Stage I machinery, the automation extends to the initial filling operation. TMR ingredients can either be loaded automatically in a stationary mixer, which facilitates mixing and shredding and fills the delivery wagons, or stored in a mixing station. Here, mechanical systems load them into auger-equipped wagons that mix the TMR before proceeding with the ration distribution.Stage III: the TMR preparation is fully automated. The AFS autonomously provides load mixing–distributing wagons with each ration component directly from bunker silos without human intervention.

Moreover, AFSs differ not only by the level of automation achieved but also by the feed delivery unit’s construction characteristics, making them suitable for the various specific housing conditions retrievable on the farm (Figure 2).

### 2.3. The Mixing Station and the Wagons

No matter the automation stage and the construction characteristics, all AFSs require a specific building section to prepare the ration: the mixing station. It is always a covered area inside or outside the barn. Here, the farmer gathers all of the ingredients to prepare the ration. The mixing station contains all of the needed equipment, which may vary depending on the AFS model and design (Figure 3), and should comply with the following requirements:(i)the temporary storage containers and concentrate dispensers must be easily accessible to make their filling operation the easiest possible;(ii)the space available must allow for cleaning and maintenance works;(iii)mixing station building characteristics should comply with hygiene and safety standards for the ingredients’ preservation.

Moreover, the building protects the TMR components from heat and direct sunlight, has good ventilation, especially during the summer, and shelters from bad weather and wildlife [68,69].

The TMR-delivering wagons may have various designs and are mainly electric-powered (Figure 3). Belt conveyor distributors represent the simplest solution for automatic TMR delivery as they bring the freshly made TMR exiting from a stationary mixer [67,69].

Wheel-driven AFSs (Figure 4a) use a track or an electrified suspended line as a direction and power source. Such wagons discharge their weight on the ground through the wheel, preventing the building structures from any potential overload.

Self-propelled wagons (Figure 4b,c,f) move throughout the barn following magnetic paths that specific magnetic sensors in the paving surface create. They may be either mixing–distributing or only distributing devices.

Rail-suspended wagons (Figure 4d,e) may mix and distribute or only distribute the ration. A rail, attached to the pillars of the barn and running all along the feeding trough, provides the suspended wagons for support, direction, and energy power.

All stage II and stage II AFSs require human intervention for bovine ration preparation because even stage II AFSs require a workload for TMR ingredient supply in the kitchen.

Fully automated models are currently in the prototype phase. They do not need the kitchen room because they can pick the ingredients from the bunker silos and provide for transport, chopping, mixing, and distributing the ration. Such systems use technologies such as LiDAR (light detection and ranging) and radar (radio detection and ranging) for free navigation within the farm site and provide for personal protection and collision avoidance [65,70].

In Italy, the situation of AFSs is very heterogeneous [71]. In 2022, there were 101 installations of robots; the self-propelled systems represent the most frequently installed devices (63%), followed by rail-suspended (26%), wheel-driven (9%), and conveyor belt (2%) models. Most AFS installations concern northern Italy, and the installed AFS typology results from considering the orography and the herd size. No stage III AFS is currently being installed in Italy.

## 3. What Farmers Perceive about AFSs

Based on the results of a survey among Italian cattle farmers [71], stage II self-propelled AFSs are more likely to equip bigger livestock-farming units. On the contrary, farmers running smaller livestock units are more prone to opt for stage I rail-suspended model AFSs, where only the chopping–mixing and ration distribution tasks are mechanized (not the mixer filling).

The robotic feeding implementation in livestock farming is quite a recent innovation. As per Lenain et al. [72], in 2020, the automatic feed-pushing technology in dairy farms was at the stage of early commercial sales. Historically, despite the need for digitalization, agriculture has been slow to react to this; such difficulty in applying innovative technologies (such as those of precision agriculture) relates to the complex decision-making problems farming operators must face, which are ascribable to the multifaceted activity of farming and to the institutional setup farmers operate in [73]. Although the diffusion rate of these robotic solutions in the Italian territory is still low, the farmers’ experience is satisfactory as most would repeat the purchase of an AFS at the same or different stage of development. As per Silvi et al. [74], user friendliness, the cost–benefit ratio, and the availability of technical assistance are the main drivers of their first choice. The cost necessary to install the system and the introduced level of automation affect the appreciation of the ease in preparing the feeding groups and the flexibility of the workload.

As expected from the recent introduction of AFSs, almost 1/3 of the surveyed farmers did not have a robotic technology for TMR cattle rationing; however, their willingness to acquire was relatively high as most of them (75%) would invest between 500 and 1500 euros per head. Such a willingness to invest relates to the desire to improve feeding accuracy and animal welfare, achieve improved production performances, reduce the workload, and introduce more flexibility in working hours, as proven by Grothman et al. [48], Pezzuolo et al. [75], and Abhijeet et al. [38], who considered the adoption of robotic feed pushers.

The reasons for adopting an AFS were corroborated by the study of Kauppinen et al. [76], who highlighted that farmers’ consideration of taking care of their well-being is, on the one hand, weakly but positively linked with animal welfare indicators, but not with increased production.

In the study of Bisaglia et al. [71], the dairy farmers who transitioned to mechanized feeding experienced significant energy savings and increases in milk production, perceiving this as resulting from the increased frequency of ration administration and feed-pushing activity by AFS wagons. Specifically, the surveyed farmers reported over 60% energy savings, a daily increase of 2.50 kg_fresh weight_ of ingested ration per cow, and a daily milk production increase of 2.94 kg per cow. According to farmers’ observations, robotically fed animals tend to spend less time waiting to consume their ration, with an increased frequency of visits to the milking area and resting activity, which benefits bovine feet and limb health, preventing cows from experiencing difficulties in walking, lying down, and standing up [77].

## 4. Discussion

The AFSs represent the last frontier of robotization of TMR rationing. Together with milking and manure management, feed automation represents a radical innovation that results in profound changes in cattle management. In Italy, its spread has occurred mainly in the regions of northern Italy where Lombardy and Trentino Alto Adige host about 60% of the plants, while Veneto and Emilia Romagna host another 30%. However, especially in central and southern Italy, the spread of technology is ongoing (Figure 5). This distribution testifies to the system’s versatility; its spread affects the most strategic Italian areas for dairy farming (e.g., hilly or lowland territories), but at the same time, it is clear how AFS technology has turned out to be meaningful even in mountain production contexts.

AFS-equipped farms have significant variability, which the described technology succeeds in modulating as it adapts very well both to already existing and newly built animal housing systems where, following the modern *holistic* design, each plant connects and integrates well with all of the others (i.e., robotic feeding, milking, and cleaning systems “communicate” with each other by exchanging information, resulting in the optimization of the efficiency of the housing system) in full compliance with good feeding practices [78].

**Figure 5 animals-13-03382-f005:**
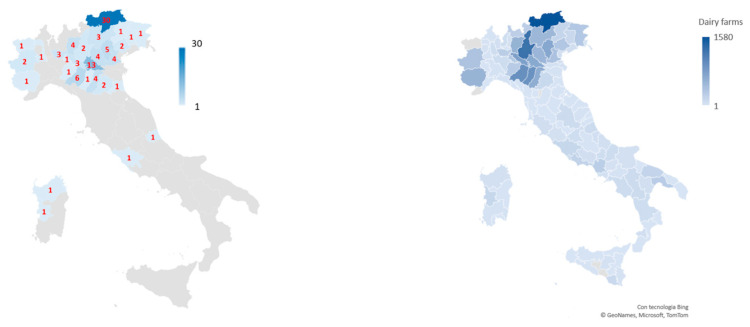
On the (**left**): territorial location of cattle farms equipped with AFS technology in Italy (numbers refer to the numbers o farms in the province). On the (**right**): territorial allocation of dairy farms (data as per the year 2021, processed from the Italian National Register of Zootechnical Statistics [79]).

### 4.1. Economic Assessment

Reducing production costs, the difficulty in finding specialized personnel and the increasing size of herds have driven the development and diffusion of specific automated solutions in the dairy and beef sector. In particular, from a technological point of view, automatic concentrate dispensers and milking robots (AMSs) have been used for several years, while the AFS has entered the market only recently.

Concerning the latter, the reasons for their success among breeders can be explained by (i) the high amount of time spent preparing and distributing the feed ration, (ii) the need to prepare a balanced and quality ration to optimize the cows’ production performance, and (iii) the possibility of feeding the ration more frequently than once/twice a day, to provide animals with fresh feed [80]. Increasing the feeding frequency increases the visits to the feeding trough over 24 h; this leads to longer feeding times and increased dry matter intake, with positive effects on cow health and production [81].

However, the adoption of an AFS results not only in an increased number of animal visits to the feeding trough but also considerably affects the organization of the daily workload, combining the reduction in labor requirements with an improvement in the quality of work if compared to the current technique based on the use of self-propelled or trailed mixing wagons. Bisaglia et al. [82] and Pezzuolo et al. [75] have shown a 50–60% reduction in the time needed to produce the total mixed ration (TMR) by switching from a conventional system consisting of a tractor-coupled mixing wagon to a stage II AFS equipped with a stationary chopping–mixing wagon positioned in the kitchen room and a rail-suspended distributor wagon.

#### 4.1.1. The Energy Requirements of TMR Operations

Conventionally, the TMR preparations rely on chopping–mixing wagons, either self-propelled or coupled to a tractor of suitable power. TMR wagons may follow three main different design architectures: (i) vertical auger wagon, (ii) horizontal auger wagon, and (iii) rotating barrel mixing wagon with a counter auger. All of these are operating machines powered by endothermic diesel engines with chopping–mixing organs driven by a mechanic or hydraulic transmission [6]. Commonly, the nominal power of the self-propelled models ranges between 88 and 210 kW, depending on the hopper loading volume. Table 1 reports the nominal power (kW) per cubic meter of capacity of the main TMR wagon types available on the European market (data provided by manufacturers, year 2022).

It should be noted that self-propelled endothermic models, equipped with drive axles usually driven by hydrostatic transmissions, require almost twice as much power per unit as towed wagons [83]. Within the represented framework, the stationary TMR wagon deserves a separate discussion; it is a recent chopping–mixing wagon with electrically powered vertical augers and a mechanical transmission. It has no feed-loading devices or propulsion organs, which makes such a TMR wagon require much less power capacity per cubic meter than traditional wagons. Such data show that no matter the mixer wagon typology, TMR preparation and distribution is a highly energy-intensive process with a substantial impact on production costs and the environment due to fuel consumption and endothermic engines’ polluting emissions [64].

The electrification of machines and plants represents a readily available solution for reducing energy consumption and environmental impact in agriculture [84,85]. An electrically operated device offers the following advantages:(a)increased control capacity: an electric actuator responds quickly and precisely, flexibly modulating the intensity and magnitude of an action;(b)greater simplicity from a mechanical point of view: greater flexibility and performance of electric drives; complete or partial elimination of noise and vibrations;(c)greater motor efficiency (95% electric vs. 38–40% endothermic diesel [86]);(d)lower energy consumption for the production process;(e)complete or partial elimination of polluting emissions;(f)improved environmental sustainability.

The electrically driven and powered (except for stage III systems) AFSs provide all of the above benefits. For varying AFS types (Section 2.1) and based on the data provided by the leading manufacturers on the European market (data referred to the year 2021), Table 2 reports a classification based on the average daily energy consumption according to the power installed. Stage I and stage II systems show a remarkable energy requirement reduction compared to stage III systems (−94%), which, as mentioned, are self-propelled, robot-driven, diesel-powered TMR wagons with all of the technical limitations already seen.

Da Borso et al. [37] observed that the daily electricity consumption of a stage II AFS, based on the installed power and the operating time, was reduced by 70% compared to the conventional tractor–TMR wagon operation. Tangorra and Calcante [64], focusing on a large dairy cow farm in northern Italy, showed a greater than 90% reduction in energy and 79% in labor required for the TMR preparation by switching from a conventional TMR wagon to a stage II AFS with a kitchen and two feeding robots. It follows that the greater energy efficiency combined with the decrease in labor from preparing the TMR using an AFS can lead to lower consumption, production costs (with savings of up to 33%), and environmental impact. Such aspects are more relevant than ever for good business management and the economic and environmental sustainability of the production process.

#### 4.1.2. Planning Issues of AFS Inclusion in a Farm

As aforementioned in Section 2, the mixing station design and the feed wagon system are the characterizing features of an AFS. Each AFS-producing company has developed technical solutions adaptable to the specific type of farm and the number of animals to be fed. Including such automated systems in new or existing buildings is a key issue; the system’s design should fit appropriately into the broader context of the design of new livestock buildings and the renovation of existing ones.

The positioning of the mixing station influences the choice of AFS: notably, the features of each cattle shed and the mixing unit’s suitability in storing and preprocessing feed ingredients are the most crucial issues to tackle during the planning phase. Moreover, the positioning of the kitchen should also consider (i) the routes of the wagon to minimize downtime and (ii) the elevation differences or slopes that may occur along such routes. These requirements may address the planning of one construction type over another.

With the introduction of these systems, when planning newly built cow sheds, it is possible to reduce the width of the foraging lane and (moderately) the number of feed bunk places with a subsequent reduction in construction costs or increase in the resting area for animals. However, the foraging lane should also be accessible to other types of mechanization to ensure the continuity of rationing in the case of automatic wagon malfunction. Furthermore, in the planning phase, the structural aspects of the building require attention, particularly when addressing the choice of suspended models. The weight of the wagon, its kinetic energy, and the resulting vibrations may preclude its installation in some shelters, especially in the case of old construction or, if recent, not having been designed according to the holistic view described above.

From an economic point of view, adopting an AFS system saves daily costs compared with a conventional feeding system (CFS) (i.e., tractor–TMR wagon) despite the greater technological level and the need for appropriate farm structures. Indeed, as per the study mentioned above [64], the daily costs of preparing and distributing a TMR with a stage II AFS are 33% lower compared to using the CFS. Calculating the mechanization costs following the ASABE standard methodology, which divides the costs of an agricultural machine into ownership and operating costs, demonstrates that the significant savings achievable in terms of labor (−79%) and required energy (−90%) allow for a rapid return on investment of adopting an AFS in the farm. These results highlight that AFSs can represent an interesting option to improve the competitiveness of farms. 

### 4.2. Animal Welfare

AFSs improve feed administration procedures, animal health, and animal welfare [88,89]. The human–animal relationship benefits livestock; however, there might be working conditions causing it to be even more damaging for the animals [90]. In such situations, using an AFS is potentially beneficial because it reduces the interactions between cattle and the stockperson.

The desirable management of the feed bunk decreases the feeding competition, resulting in negative animal interactions (e.g., pushing, head butting). In particular, the frequent passage of wagons suggests to animals that there is continuous availability of feed, inhibiting the aggressiveness of dominant animals and relaxing the submissive ones, with a subsequent increase in animals’ voluntary ingestion over 24 h [89]. Schneider et al. [91] assessed some parameters of fattening bulls fed with an AFS (e.g., body condition score, individual behavioral observations, and carcass weights), pointing out the significance of providing animals with a TMR six times in 24 h.

Assessing cattle behavior involves observing several activities at different moments of the day (e.g., when resting in cubicles, milking parlor, and watering places). Concerning the feeding stage specifically, Brito et al. [89] recognized a dual role of the AFSs: firstly, the AFSs improve dairy cattle welfare; secondly, the AFSs can also record many animal behaviors according to Foris et al. [92], who found the AFSs helpful to assess the dominance relationship in dairy cattle.

To ensure adequate access to the feed and to prevent the animals from competing with each other, guaranteeing a desirable ratio between herd size and the feed bunk length is also fundamental. In addition, an optimal feed bunk per head is strategic to ensure a good animal welfare standard. However, the kind of feed bunk, the type of feeding, the feed distribution system, and the feed management are among the factors that, in turn, affect the optimal number of available places [93].

Currently, there are two types of feed bunks: (i) those with delimited places, equipped with self-capture systems (either with or without anti-suffocation systems), and (ii) those with two horizontal tubes, commonly used for fattening cattle. The first one, albeit of simple technology, allows routine operations on dairy cows and, therefore, requires the number of available feed bunks to match the number of housed animals [94]. Lactating cows, characterized by a longer productive life, require careful attention and feeding, which the single manger ensures. On the other hand, the shorter housing of fattening cattle does not require obligatory spaces in the feed bunk as treatments on a single fattening animal do not occur frequently. According to our best knowledge, on the one hand, AFSs are rarely used in beef systems, despite their technology meeting the needs of fattening cattle farms; on the other, their occurrence in dairy cattle farms is becoming quite widespread, although not ubiquitous [73].

Concerning feeding administration, in the case of simultaneous feeding, the number of optimum places should match 100% of the animals because all of them go to the feed bunk simultaneously at the time of distribution. It lowers to 70% of the reared animals in the case of continuous feeding because the barn’s design considers the presence of a feeder that makes the feed available 18 h per day, causing animals to alternate to access the feed bunk. 

The feed distribution system and management play a crucial role as feeding is a predominant behavior in dairy cattle; dairy cows spend 3 to 5 h day^−1^ feeding, consuming 9 to 14 meals daily [95]. Traditionally, housed dairy cows are provided with fresh feed twice or only once daily, mechanically. In this case, animals receive small quantities of fresh feed several times daily. Distributing feed in several daily meals limits competition at the feed bunk and improves the digestive function of each head [96], confirming the findings of Robles et al. [97], who highlighted that a high feeding frequency improves the ruminal condition and the digestive performances [97].

Properly managing the feed at the feed bunk is a further important issue. Dairy cows tend to discard/sort feed in search of the most palatable ingredients, pushing it away from the feeder [53]. For this reason, it is necessary to push it closer again. When performed automatically by feed-pusher-type robotic units, the increase in the daily feed approach has been associated with increased milk yield [98]. Studying farms equipped with automatic milking systems, Siewert et al. [99], Deming et al. [100], and King et al. [101] highlighted the benefits of increased feed-pushing frequency, which resulted from improved feed access, decreased sorting, and less time spent searching for feed and increased time spent lying down, which all have positive effects on milk yield.

A recent investigation confirmed that the frequency of feed distribution and feed pushing could reduce and limit agonistic interactions at the feed bunk [102]. When animals receive fresh feed once a day, most of the cows go to the feed bunk as soon as the feed is delivered; increasing the number of feed distributions results in decreased feed bunk attendance (Figure 6) at feed distribution and pushing back. Agonistic interaction frequency has been higher in conventional feeding farms than in AFS-equipped ones (Figure 7).

The introduction of automatic systems for feed distribution and feed approach can, on the one hand, reduce and make the work of the farmer flexible [37] and, on the other hand, increase the level of animal welfare by reducing food competition in the feed bunk and the stress of the animals, thanks to the possibility of distributing and approaching the feed ration several times a day.

## 5. Conclusions

AFSs represent the evolution of the TMR preparation technology with subsequent advantages regarding workload, animal nutrition, and animal welfare optimization. The market offers farmers quite a wide choice of machinery that can adapt to the animals’ needs and the constraints resulting from the farming site’s peculiarities.

However, the reviewed studies and farmers’ perceptions pinpoint that when choosing the AFS, farmers should holistically consider the more appropriate feed-delivering technology and structural and organizational aspects of the shed.

In particular, the chosen feeding system should fully match the farm’s needs, being able to cope with the needs of the various groups of animals (all must receive the same amount of feed; the system components should guarantee adequate autonomy) and the operators’ needs (suitable loading and unloading solutions; provision of an alternative feeding method in case of equipment failure).

Concerning the cow shed, the AFS arrangement should match the farm work cycles, and its placement (external, eaves side, or gable side development) should consider the path the wagons follow and the need for protection that the kitchen room requires to prevent TMR components from deterioration. Already in the planning phase, the paving of the barn needs accurate consideration. Whether newly built or not, the paving of the shed must have an appropriate design (suitable slope, smooth surface, silage effluent, and cleaning water draining systems). The building structure (particularly its load-bearing capacity) and the feed alley width should match the characteristics of the AFS to achieve maximum efficiency from the machine. 

AFSs are easy to use, making them fundamental for the involvement of younger generations in animal farming and increasing the inclusivity of such an activity with the subsequent fostering of female entrepreneurship. However, there are also some warnings: Running an AFS does not free farmers from complying with good feeding procedures because the ingredients’ quality and the management’s appropriateness affect the feed quality and animal production (Figure 8). Healthy animals fed balanced diets and provided with abundant supplies of fresh water are the most productive, the most profitable to the farmer, and the most efficient users of nutrients.

## 6. Future Directions

The McKinsey Institute [103] estimated that a high percentage of working time in farming (more than 50% on average) relates to repetitive, physical activities performed continuously with little decision making, making even the partial automation and robotization of many agricultural and barn operations very interesting and, perhaps, unavoidable.

Fostering automation and robotization in animal farming goes far beyond the technical aspects. On the one hand, robotic milking and automatic TMR feeding are already established technologies currently undergoing further consolidation in many advanced countries; on the other, keeping the focus on feeding automation, AFSs represent an excellent opportunity for the new generations of farmers (more prone towards digital technologies) to continue farming even in challenging territorial contexts. Whether totally or partially automated, such feeding systems are a valid option for farmers, provided their integration into equally modern and advanced farm management occurs.

A further aspect favoring automation concerns energy use—particularly for the highly efficient electric engines—which in many cases can be self-produced by the farm itself, increasing the environmental sustainability of livestock activities. 

Considering the economic aspects, it should be noted that granting subsidies plays an essential role in pushing farmers to adopt the most modern digital technologies, including AFSs and the related costs of introducing them into existing stables.

Moreover, automatic systems improve animal welfare by reducing conflicts surrounding access to the feeder and providing farmers with important information on animals’ physiological state thanks to the numerous interconnected sensors available. This latter aspect, however, requires farmers to be increasingly inclined to integrate their skills and experience with the objective information resulting from modern digital systems.

## Figures and Tables

**Figure 1 animals-13-03382-f001:**
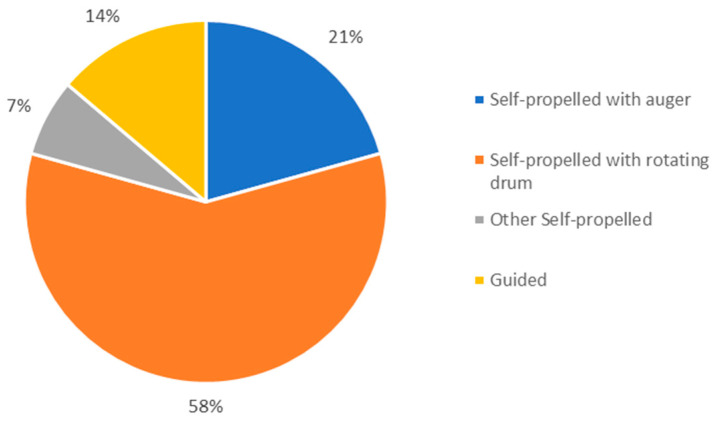
Models and automatic feed pusher classification (data by CREA, 2021 [63]).

**Figure 2 animals-13-03382-f002:**
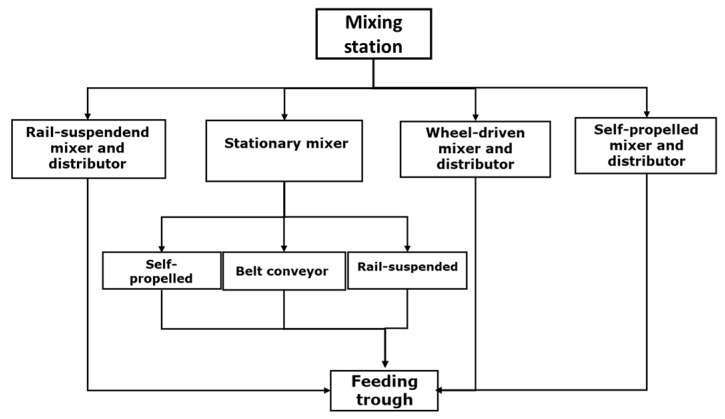
Models of AFSs on the market (from Haidn, 2014 [67]; modified by CREA).

**Figure 3 animals-13-03382-f003:**
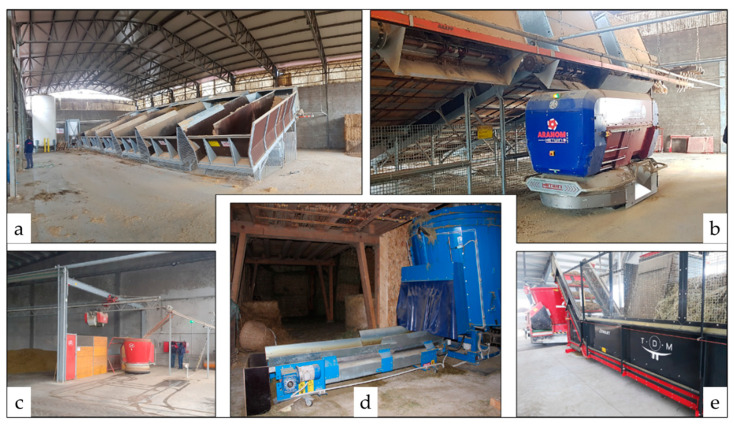
Different types of kitchens. Kitchen with temporary storage containers (**a**,**b**,**e**); self-loading crane experience (**c**); fixed mixing with belt conveyors (**d**). Pictures taken by the authors.

**Figure 4 animals-13-03382-f004:**
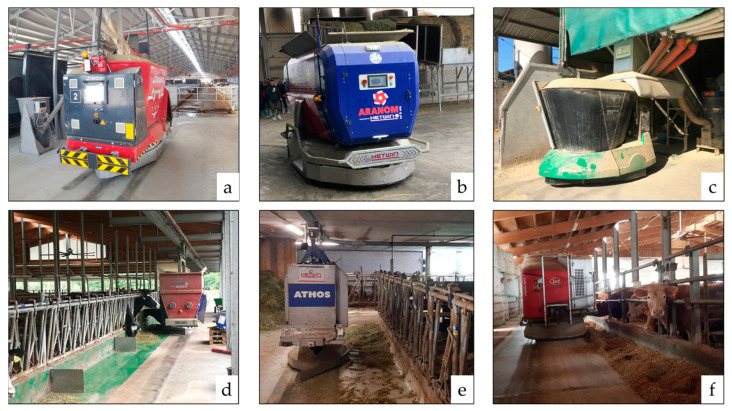
Different models of AFS wagons: wheel-driven (**a**); self-propelled (**b**,**c**,**f**); rail-suspended chopper-mixing and distributor (**d**); rail-suspended distributor (**e**). Pictures taken by the authors.

**Figure 6 animals-13-03382-f006:**
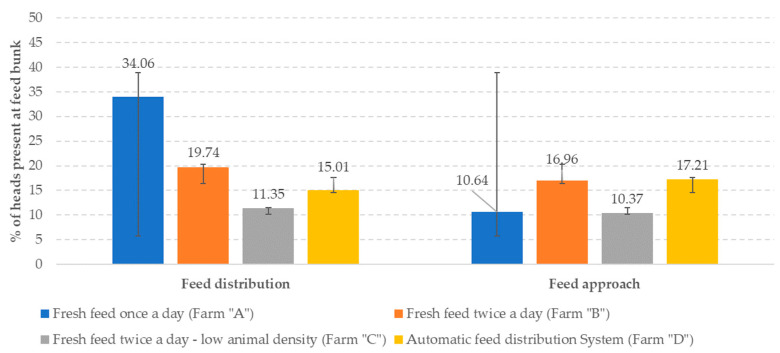
Feed bunk attendance at varying feed distribution frequency (from Motta et al. [102]).

**Figure 7 animals-13-03382-f007:**
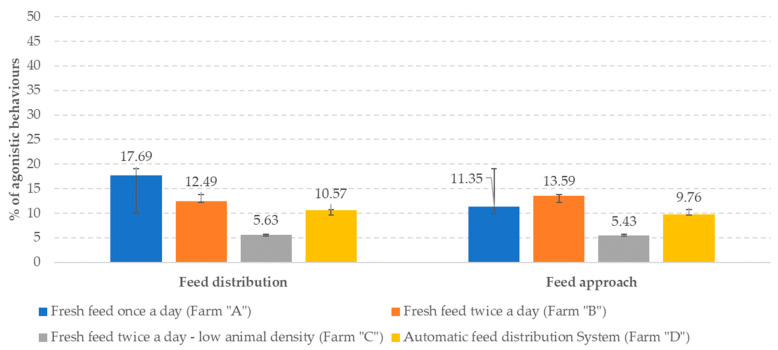
Antagonistic approaches at varying feed distribution frequency (from Motta et al. [102]).

**Figure 8 animals-13-03382-f008:**
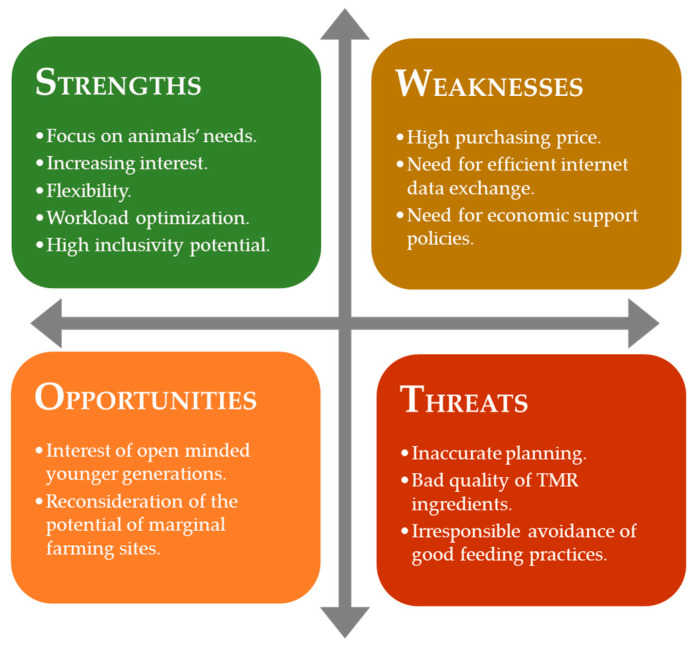
Result of assessing AFS position (SWOT analysis) at the current diffusion stage.

**Table 1 animals-13-03382-t001:** Nominal power (kW) per cubic meter of the capacity of the main TMR wagon types available on the European market (data provided by manufacturers, year 2022).

Typology of Mixer Wagon	Type of Engine	Average Power Absorption for Unit Capacity (kW m^−3^)
Self-propelled vertical mixer	Internal combustion engine	7.3
Self-propelled horizontal mixer		8.1
Rotating cylinder mixer		7.1
Trailed vertical mixer		3.4
Trailed horizontal mixer		4.4
Stationary	Electric engine	1.4

**Table 2 animals-13-03382-t002:** Average daily energy absorption of different AFS types [87].

Type of AFS	Energetic Absorption (kWh/day)
Stage I	20–45
Stage II	30–35
Stage III	570 *

* AFSs powered by a diesel internal combustion engine.

## Data Availability

The data presented in this study are available on request from the corresponding author.

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
