# Peer review of "Increased Cattle Feeding Precision from Automatic Feeding Systems: Considerations on Technology Spread and Farm Level Perceived Advantages in Italy"

_animals, 2023, doi:10.3390/ani13213382_

Round 1

Reviewer 1 Report

Comments and Suggestions for Authors

Please include how the reduced work loads is converted into profit for farmers. Energy saved vs profit generated etc. Also provide the feasibility of the implementation on wide range of farms in terms of size. How small farmers benefit from this if they have small farms? How the feed bunks are connected in this entire robotic process? How often we need to feed the cattle in a day? and how robotic process could help with that?

Comments on the Quality of English Language

Inconsistent use of abbreviations: The text uses both "Automatic Feed Pushers" (AFPs) and "Automatic feed pushers" without consistency in capitalization. some sentences are quite long and cloud be broken into shorter more concise sentences for improved readability. Clarity issue such as the sentence Figure reports the market offer of automatic feed pushers could be clear. Also provide what are the methods available to identify animal intakes?

Author Response

I am attaching the detailed responses to your remarks.
Regards

Reviewer 2 Report

Comments and Suggestions for Authors

Overall comments

1.       The structure of the paper is very odd. It is not a Review, it contains a mix of experimental (survey) results and adds a section of Case study data from individual farms in a Supplementary section. Some of the sections are written more as Farm Advisory publications using terms such as ‘should…’ to describe farm management issues in terms of how farmers should use the technology.  It further displays full results (including as histograms), but without methodology etc, from other authors, the source of which does not appear to be peer-reviewed (see 5 below).

2.       The experimental results from survey appear without detail of methods, these need full inclusion of the survey wording (in a Supplementary is fine), but also clearer title legends and descriptions of axis legends of any graphics. Most social science publications require the full wording of any questionnaires or surveys and this should be the case here.  For example, without the survey wording it is impossible to understand to what precise parameter each of the individual histograms in Figure 6 refers. Figure 10 plots each farm (each dot) as a continuous variable, mainly appearing as a cluster, but again the labels ‘Score’ , ‘Interest’ are completely unclear without better explanation in legends but also access to the survey form.  Data tables are reccommended.

3.       Overall major findings are not well justified. Line 491 ‘AFS ….improve animal welfare’ and the Abstract and Simple Summary say also improvements in animal welfare (though the including sentences do mention farmer perspectives). Figure 6 does not clearly show numbers of farmers who gave different 1-6 scores and Results for this important ‘Result’ do not seem to be anywhere else. Moreover, it’s a challenge to compare scores from the main survey questions against a financial parameter (unexplained). The statement ‘all means differ significantly (p<0.001)’ is the only indication given of distributional variation in the answer to different questions, but given the relatively small sample sizes this statement is barely credible.

4.       The standard of English is overall quite good. But there are many minor oddities in phraseology and words that are probably typos, but not picked up unless checked by an English speaker (same and some example). Not a big thing but the word ‘Kitchen’ is heavily used for the feed preparation facility. I cannot imagine any British, Irish, North American farmer calling it this. I would use ‘mixing station’ or similar. I have provided a ‘red-pen’ pdf to identify some of examples of where the English needs some checking, and editing. Not a full review of every typo.

5.       As above, I do not think it is valid for a peer reviewed publication to include the Case Study data and results within a Supplementary section. Data presented needs clear description of methods within a main paper Methods section. The supplementary section does not seem to perform much of a function in informing the main paper. I would suggest full removal.

6.       The publication of the results from Motta, as a series of histograms (figures 12/13) within your paper needs fuller scrutiny. If to be included in this way, then full methodology should be provided as with more formal results. It appears that this data is from a non-peer-reviewed article. It has one treatment per farm, and this farm and ‘system/treatment’ are fully confounded, and I do not believe acceptable for publication within a peer review paper.   

Detailed comments by section or line numbers

Title: I would suggest shortening title to start at ‘Automatic….’. because increased precision is not really covered in paper.

L31: the first line of the abstract is important. However, this sentence re climate change and adapting to its (Climate Change) financial impacts is not really covered in the main narrative.

L60. “..almost all cattle farms adopt it”. Can this have a reference? And a context, Italy, Europe? I do not believe ‘almost all’ would cover beef cattle farms in semi-extensive systems in the UK and Ireland. It might be truer of dairy farms.

 L164  “…the pavement” . This is not a term I would hear on British farms. ‘feed passage’ would be most commonly used.

L176 ‘1250 robots’ I’m not sure what this means, but think it might be better as ‘1250 robotic units on farms’, please review

Figure 2 and L196 onwards . The Kitchen. Feed mixing area, feed mixing station, feed preparation area? All sound better. British farmers might refer to the mixing shed if indoors. Feed mixing station might be best in English for a global readership.

L268 As you have an English version, the English version should be provided in a Supplementary.

L278 and 290 onwards. Wording for completion of survey needs to be precise and clear. So better to say ‘surveyed’ rather than ‘interviewed’ – when it covers all farmers who filled in the q’aire, by different mechansims. ‘Interviewed’ should only be used when it refers to people who answered ‘face to face’ questions.

L294+ The standard results should be presented in a table. Figure 6 does not provide any counts, including tabulation of regression statistics and any statistical differences between means. ‘All highly significant’(L296-297) should have more information on what test was used, putting p values in a table too makes sense.

Figure 6 as noted earlier – please review carefully. The value of graphing the Survey score against the financial value is unclear. Its unclear the value of plotting these two elements together as the text does make much of any relationship. As before providing a simple table might be preferable

L336 ‘national territory’ – if this means Italy, suggest use Italy

Figure 10. These graphs need careful review. The size of dots appear all to be the same but text (L346) says they are different for herd sizes.

Both Score and High Med Low are plotted as continuous variables, though clustered, but narrative at least for Scores is that they are integers, categorical. The colours refer to energy. Having  all these 4 variables on one graphic might be too confusing.

L347 states ‘what can be deduced from the graph…’ but none of the things mentioned seem easily apparent.

L362 30% and 60% . Are these survey figures or some other statistics?

Table 2 – source of data could be supplied.

L490 4.2 Animal Welfare. This section reads like an advisory/extension document. More references are needed to support the statements given.

L590 Some of the Conclusions appear without any mention in earlier Results or Discussion -  for example the comments about this technology fostering female entrepreneurship, rather an odd statement, without evidence.

L618 The final paragraph again provides statements which have had little evidence provided in the paper. The animal welfare benefit is again noted without clarity of what the survey data shows.

Supplementary Case Study

No comments made about this section – other than it does not fit and I recommend it should not included in this manner.

Comments on the Quality of English Language

see attached pdf file for some 'English' comments. 

Author Response

Dear Sir or Madam,

Please find attached the file containing the detailed responses to your remarks. 

Regards

Reviewer 3 Report

Comments and Suggestions for Authors

It is basically an interesting topic. However, I am somewhat lacking the scientific novelties. Many of the points mentioned are known from other publications for a long time.
In the investigation should also be addressed to the AFP and AFS differentiated in the literature. In the results we talk mainly about AFS.
In material the explanation of the questionnaire is missing. What was it designed for? What else was asked? When did the survey take place?
There is a clear lack of depth in the results. Analysis of the surveyed farms, farm managers, size, structure and comparison to the average are missing.
Nothing is said about the AFSs surveyed. Here a detailed analysis of the survey results is missing.
In some cases, the results jump between different populations. How many farms are there in which statement?
Figures 7 and 8 can be made simpler and thus better understandable .
The agglomerations in Fig. 9 are not discussed. Here the explanation is missing.
The four areas of investigation - labor, costs, health and energy - are only superficially analyzed.
The entire work lacks scientific depth and new insights.

Author Response

Dear Sir or madam,

Here you are in attachment the file containing the responses to your remarks.

Regards

Reviewer 4 Report

Comments and Suggestions for Authors

The article appears to be of significant importance to the field of agricultural research, especially in the context of automation. It provides a multifaceted analysis of AFS, taking into account economic, environmental, and animal welfare factors. Such a comprehensive study can guide future farming practices, influence policy decisions, and shape the development of next-generation farming technologies. The research in question involved data collection from farmers via questionnaires. Even if it might seem benign, there are still ethical considerations involved, such as ensuring participants gave informed consent and that their data are kept confidential.

 The authors mention that data was collected in person, over the phone, and online. Each method has its own ethical implications. For instance, ensuring online participants have given informed consent might be more challenging.

 Even though the research might seem low-risk, without a proper review by an ethics committee, we can't be sure that all potential risks were considered and mitigated.

 It's crucial for the authors to clarify the absence of mention of ethics committee approval. If approval was sought but simply not mentioned, they should rectify the record. If it was not sought, they should clearly justify why they believed it wasn't necessary and consider seeking a retroactive ethical review or acknowledge this as a limitation in their findings.  

Author Response

Dear Sir,

Thank you for your appreciation. 

We'll consider the ethical review for future revisions of the manuscript.

Kind regards

Round 2

Reviewer 3 Report

Comments and Suggestions for Authors

The text has gained even more quality through the revision.